# The relationship between spatial configuration and functional connectivity of brain regions revisited

Janine Diane Bijsterbosch[1]*, Christian F Beckmann[2], Mark W Woolrich[3], Stephen M Smith[1†], Samuel J Harrison[1†]

[1]Centre for Functional MRI of the Brain (FMRIB), Wellcome Centre for Integrative Neuroimaging, Nuffield Department of Clinical Neurosciences, University of Oxford, Oxford, United Kingdom; [2]Donders Institute, Department of Cognitive Neurosciences, Radboud University Medical Centre, Nijmegen, Netherlands; [3]Centre for Human Brain Activity (OHBA), Wellcome Centre for Integrative Neuroimaging, Department of Psychiatry, University of Oxford, Oxford, United Kingdom

**Abstract** Previously we showed that network-based modelling of brain connectivity interacts strongly with the shape and exact location of brain regions, such that cross-subject variations in the spatial configuration of functional brain regions are being interpreted as changes in functional connectivity (Bijsterbosch et al., 2018). Here we show that these spatial effects on connectivity estimates actually occur as a result of spatial overlap between brain networks. This is shown to systematically bias connectivity estimates obtained from group spatial ICA followed by dual regression. We introduce an extended method that addresses the bias and achieves more accurate connectivity estimates.

DOI: https://doi.org/10.7554/eLife.44890.001

*For correspondence:
janine.bijsterbosch@wustl.edu

†These authors contributed equally to this work

**Competing interests:** The authors declare that no competing interests exist.

**Reviewing editor:** Chris Honey,

## Introduction

Resting state functional magnetic resonance imaging (rfMRI) can be used to characterise the rich intrinsic functional organisation of the brain (*Biswal et al., 1995*). Distributed networks of brain regions exhibit coordinated activity of neural populations both within and between networks, known as functional connectivity (*Fox et al., 2005*; *Greicius et al., 2003*; *Raichle et al., 2001*). Analytical approaches to functional connectivity can be broadly split into voxel-based methods (deriving map-based connectivity estimates to study the spatial organisation of networks) and node-based methods (approaches based on network-science that describe connectivity in terms of 'edges' between functional brain regions) (*Bijsterbosch et al., 2017*; *Rubinov and Sporns, 2010*). Here, we focus on node-based methods and ask how complex aspects of spatial organisation may influence estimated node-based functional connectivity.

There is growing interest in interindividual differences in these node-based functional connectivity patterns and their potential use as markers for health and disease (*Finn et al., 2015*; *Kaiser et al., 2015*). We previously showed that interindividual variability in estimated functional connectivity between brain regions is, to a large extent, driven by variability in the spatial organisation (i.e., the precise shape and location) of large-scale brain networks (*Bijsterbosch et al., 2018*). Specifically, our previous results revealed that node-based functional connectivity as normally estimated from rfMRI data is influenced by a mixture of spatial and temporal factors, with spatial information explaining up to 62% of interindividual variance. This is unexpected and arguably undesirable, because temporal correlation-based functional connectivity estimates are often considered as an

accurate representation of temporal coupling strength between neural populations. Therefore, there is a need to better separate interindividual variability in spatial and temporal domains in analytical approaches to rfMRI data, in order to gain a better understanding of interindividual connectivity profiles and derive more interpretable associations with behaviour. To identify connectivity measures with the strongest unique association with behaviour (i.e., potential measures for clinical biomarkers), we need to understand and address the reason for the previously reported conflation of temporal and spatial sources of interindividual variability (*Bijsterbosch et al., 2018*).

In our earlier work we tested a number of parcellation methods (i.e., voxel-based methods used to parcellate the brain into a small set of large-scale functional networks that consists of brain regions that exhibit similarities in their BOLD timeseries, or into a larger set of contiguous regions that consist of interconnected voxels). We tested a group-level functionally-defined hard parcellation (*Yeo et al., 2011*), an individualised multimodal hard parcellation (*Glasser et al., 2016*), and both high and low dimensional soft parcellations estimated with Independent Component Analysis (ICA) (*Beckmann and Smith, 2004*). We found that the cross-subject variability in spatial organisation strongly influenced node-based functional connectivity estimates regardless of the parcellation type/ method (*Bijsterbosch et al., 2018*). There are a number of possible explanations for this previously observed influence of spatial organisation on functional connectivity estimates, as discussed below.

Firstly, it is possible that our previous findings were driven by misalignment between the spatial boundaries of functional regions ('nodes') estimated at the group-level, and the true spatial organisation in an individual subject. This type of misalignment refers to the lack of voxel-to-voxel correspondence across individuals that remains even after applying standard alignment approaches (here surface-based multimodal alignment [*Robinson et al., 2014*]). A direct result of such misalignment, if not accounted for in the methodological approach used to estimate node timeseries, is that those timeseries will incorrectly contain a mixture of multiple 'true' node timeseries. The effect of such mixed timeseries is potentially profound, leading to a sharp reduction in the accuracy of subsequently estimated temporal correlations ('edges') (*Smith et al., 2011*). Parcellations are typically defined at the group level in order to ensure correspondence between nodes across subject, although some parcellation approaches described above include steps to account for misalignment (e.g. using a classifier for the multimodal parcellation, and dual regression for ICA [*Hacker et al., 2013*; *Nickerson et al., 2017*]). While these approaches are expected to appropriately account for misalignment, our previous findings suggested that spatial information strongly influenced estimated temporal correlations despite the use of dual regression or classifier steps to obtain subject specific maps (*Bijsterbosch et al., 2018*).

Secondly, it is expected that different parcellation methods preferentially represent connectivity information in either the temporal or spatial domain. For example, 'nodes' in a low dimensional soft parcellation take the form of spatially extended networks of brain regions, and therefore the connectivity between functional regions that are included in the same network are described in the spatial map. Conversely, these functional regions would be split into separate nodes when using a high-dimensional (contiguous) hard parcellation, and consequently the same connectivity between functional regions must be described in the edges. In our earlier work, we tested to what extent our findings were explained by this type of within-network connectivity information that may be represented in low-dimensional spatial maps. The findings showed that the direct mapping between low-dimensional extended networks and high-dimensional node edges does not explain the influence of cross-subject variability in spatial organisation on cross-subject variability in edge estimates (see Supplementary file 1e in *Bijsterbosch et al., 2018*).

Thirdly, it is possible that the assumptions that underlie the estimation of the group-level parcellation are incorrect (e.g. the spatial independence constraint in spatial ICA). If the assumption of spatial independence is incorrect, this would result in mis-estimated group maps, which may affect how functional connectivity is represented in downstream dual regression estimates. Breaking the spatial independence assumption implies the presence of relatively extensive amounts of spatial overlap between nodes, such that spatial correlations between node maps are present. While this type of complex, overlapping spatial structure does not easily fit with the intuitive notion of nodes (and is precluded, by definition, in hard parcellations), the example of overlapping receptive fields with

selectivity for stimulus orientation, length, width, or colour in V1 demonstrates the possibility for hierarchically overlapping functional systems (*Van Essen and Maunsell, 1983*). Indeed, task-based functional activation patterns can be more accurately captured based on soft-parcellations with inherent scope for overlap, compared with hard parcellations (*Bzdok et al., 2016*). It is possible that spatial ICA may underestimate spatial overlap between nodes (as a result of enforcing spatial independence in the estimated maps), although previous work has shown that, in the presence of noise, this overlap can be recovered using thresholding (*Beckmann et al., 2005*).

An alternative parcellation method, designed to avoid the spatial independence constraint, is PROFUMO. This adopts a hierarchical Bayesian framework that includes spatial priors (for map sparsity and group map regularisation) and temporal priors (consistent with the hemodynamic response function) (*Harrison et al., 2019*; *Harrison et al., 2015*). The Probabilistic Functional Mode (PFM) maps obtained from PROFUMO commonly show relatively extensive amounts of spatial overlap (and hence spatial correlation) between nodes. These overlap regions may contain a spatial representation of complex between-node patterns of functional connectivity. Therefore, it is possible that the presence of these spatial correlations in PFM maps, which are not present in un-thresholded group spatial ICA maps (by design, as a result of spatial independence), may explain our previous results (*Bijsterbosch et al., 2018*).

The aim of this work is to disambiguate functional connectivity information in the temporal and spatial domains, and to determine the influence of different algorithms used for parcellation and connectivity estimation. Specifically, we focus on soft parcellations to determine the interaction between complex functional brain organisation and the estimation of spatial nodes and temporal edges. We adopt simulation approaches that allow direct control and knowledge of the ground truth, provide mathematical explanations, show examples observed in real data, and test associations with behaviour.

## Results

### Dual regression performance in the presence of misalignment

One potential explanation for our previously reported findings (*Bijsterbosch et al., 2018*), is spatial misalignment between the exact location of node boundaries in individual subjects compared with group maps. In a typical ICA pipeline, group ICA (using the temporal concatenation approach) is followed by dual regression (*Nickerson et al., 2017*). Dual regression aims to estimate subject-specific spatial maps that accurately capture spatial organisation, accounting for any misalignment with the group maps. We will refer to group ICA followed by dual regression as the 'ICA-DR' pipeline throughout.

To assess the degree to which the ICA-DR pipeline is affected by spatial misalignment, we make use of the fact that data from the Human Connectome Project are available using two different surface-based versions of multimodal surface matching (MSM) spatial alignment (*Robinson et al., 2018*; *Robinson et al., 2014*), and as volumetrically aligned data. Cortical alignment using 'MSMSulc' is driven by gyral folding patterns alone, whereas cortical alignment using 'MSMAll' incorporates multiple spatial features including folding patterns, myelin maps, functional resting state networks, and visuotopic maps for alignment. Previous work has shown substantial improvements in alignment when using MSMAll compared with MSMSulc, and both types of surface alignment are superior compared with volumetric alignment (*Coalson et al., 2018*).

Here, we directly compare spatial alignment using MSMAll, MSMSulc, and volumetric data at the single subject level. Firstly, we perform single-subject ICA independently on each dataset (temporally concatenated across four runs; dimensionality = 25). Given that the data are identical apart from the alignment procedure, any differences in the spatial network boundaries extracted with ICA from the MSMAll or MSMSulc data should be driven exclusively by misalignment (although note that for volumetric data other tissue types such as white matter and CSF may influence the network decomposition). Furthermore, these single-subject ICA results are not influenced by other subjects or by the group, and can therefore be viewed as the 'ground truth' spatial organisation in this subject for each respective alignment type. Secondly, we perform dual regression against group maps obtained from 1004 subjects aligned using MSMAll (dimensionality = 25). Matching group maps in volumetric space were obtained by regressing each subject's MSMAll stage one dual regression

timeseries into that subject's volumetric data, these maps were then averaged across all subjects and entered into a spatial ICA with the same dimensionality (to ensure spatial independence, which may not be fully preserved in the regression and averaging steps). These group maps are expected to better represent the subject-specific MSMAll-aligned data compared with the subject-specific MSMSulc and volumetrically aligned data. Therefore, we can directly test how well dual regression captures the 'ground truth' spatial organisation obtained using single-subject ICA for each respective alignment type. This procedure was repeated separately in a subset of N = 22 subjects.

Firstly, we perform spatial correlations between group maps and single-subject spatial ICA maps to determine how well the group maps represent the subject-specific organisation. Correlations are transformed using Fisher's R-to-Z and entered into a one-way ANOVA with a factor for alignment (3 levels corresponding to MSMAll, MSMSulc, and volumetric). The main effect was significant (F(2,1098)=673.2, p<10$^{-10}$), and all post-hoc paired tests were significant after Bonferroni correction (MSMAll-MSMSulc $\Delta Z$=0.045, p<10$^{-10}$; MSMSulc-volumetric $\Delta Z$=0.152, p<10$^{-10}$; see *Figure 1*, *Figure 1—figure Supplement 1A*).

Next, we estimate the correlation between subject-specific maps obtained with dual regression and "ground truth" maps, separately for MSMSulc, MSMAll, and volumetrically aligned data. Results from a one-way ANOVA with a factor for alignment (3 levels corresponding to MSMAll, MSMSulc, and volumetric), showed a significant main effect, F(2,1098)=842.2, p<10$^{-10}$. This effect was driven by a significantly lower correlation between dual regression and "ground truth" maps in volumetric data (MSMSulc-volumetric $\Delta Z$=0.242, p<10$^{-10}$). Conversely, the difference in this correlation between MSMAll and MSMSulc did not reach significance ($\Delta Z$=0.013, p=0.061; see *Figure 1*, *Figure 1—figure Supplement 1B*).These results show that, despite misalignment between MSMSulc subject data and MSMAll group maps, dual regression was largely able to overcome this misalignment to estimate the subject-specific spatial organisation. However, dual regression was not able to correct for the more substantial amounts of misalignment observed in volumetrically aligned data.

An example of an individual ICA component is shown in *Figure 1*. These results qualitatively illustrate the extent to which dual regression corrected for minor misalignment between MSMAll and MSMSulc. There was a clear shift in the parietal node of the default mode network in MSMSulc data (*Figure 1C*) compared with MSMAll data (*Figure 1A*). Nevertheless, dual regression against the identical set of group maps is able to accurately estimate the shifted versions of the subject maps (*Figure 1B and D*).

## Cross-subject relationship between temporal and spatial connectivity

As discussed in the introduction, there are multiple potential reasons for the previously observed influence of spatial information on estimated node-based functional connectivity, including spatial misalignment and inappropriate parcellation assumptions. In the previous section we show that the contribution of spatial misalignment is likely to be relatively minor provided that surface-based alignment is used in conjunction with the ICA-DR pipeline. Next, we aim to test the role of inappropriate parcellation assumptions.

Previous rfMRI research has utilised the spatial shape and amplitude of intrinsic networks (*Filippini et al., 2009*), or temporal correlation patterns between timeseries extracted from nodes (*Smith et al., 2011*). However, correlations between network/node spatial maps ("spatial edges") are not commonly studied. A potential reason for this is that many parcellation approaches (by design) result in binary, non-overlapping node maps (*Glasser et al., 2016*; *Gordon et al., 2016*; *Yeo et al., 2011*). For the purposes of this paper, we define temporal edges as the correlation between fMRI timeseries extracted from two separate nodes, i.e., $\frac{cov(T_X,T_Y)}{\sigma_{TX}\,\sigma_{TY}}$, where $T_X$ and $T_Y$ are vectors (size timepoints x 1) representing the extracted timeseries of two different nodes ($X$ and $Y$; *Figure 2*). Spatial edges are mathematically defined as the correlation between the two node spatial maps, i.e., $\frac{cov(S_X,S_Y)}{\sigma_{SX}\,\sigma_{SY}}$ (*Figure 2*). In the case of grayordinate data, $S_X$ and $S_Y$ (the spatial maps of nodes $X$ and $Y$) are represented as vectors (size 91282 x 1), but three dimensional (volumetric) spatial node maps can also simply be vectorised to allow straightforward calculation of the correlation coefficient. To begin to elucidate the interaction between estimated functional connectivity in the spatial and temporal domains, we first assess the relationship between temporal correlations and spatial correlations estimated with spatial ICA using HCP data.

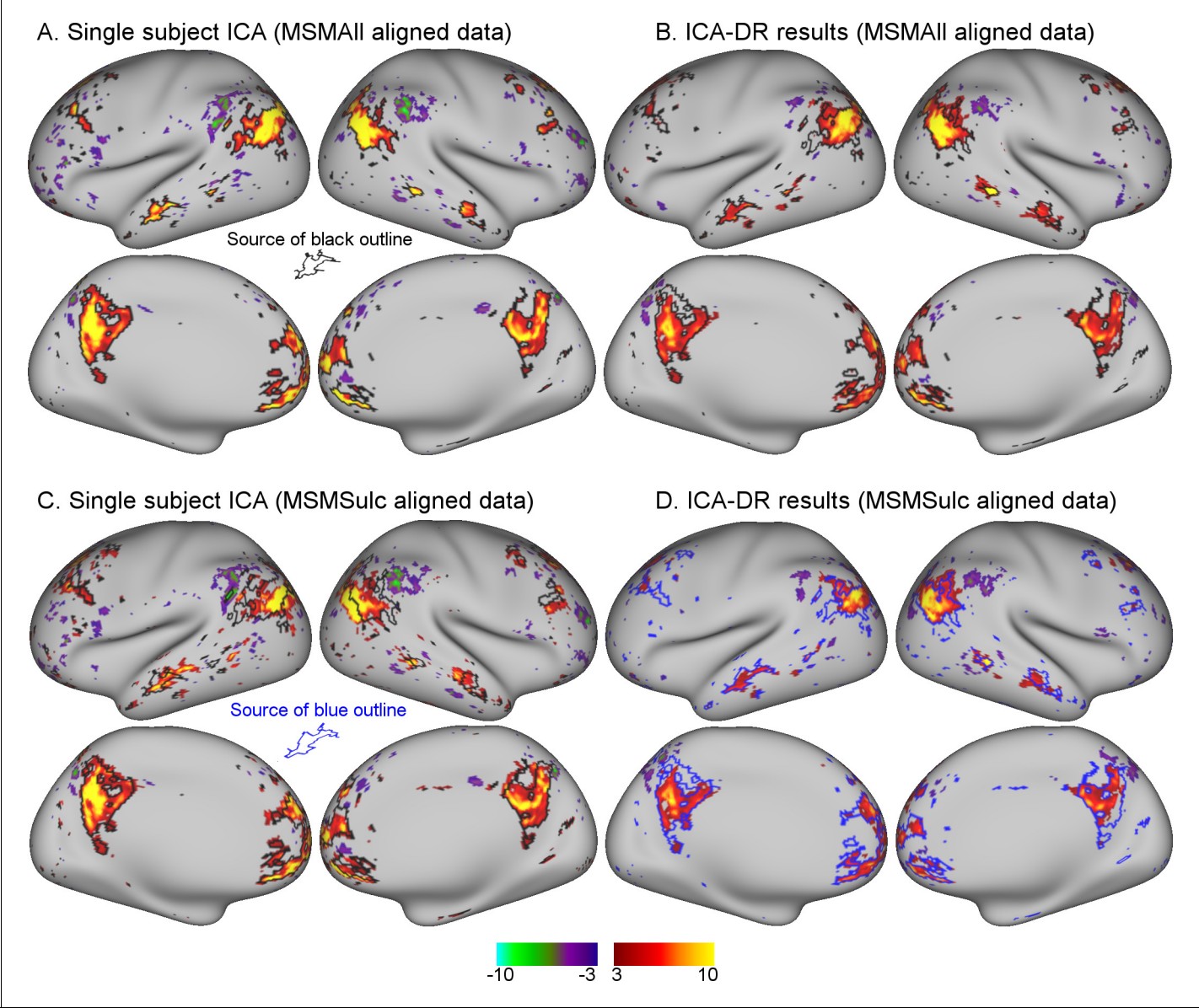

**Figure 1.** Comparison between MSMAll and MSMSulc alignment on dual regression results from an individual subject. (A) Single-subject ICA result from MSMAll data acts as 'ground truth'. (B) Dual regression of this subject's MSMAll data against MSMAll group maps captures subject-specific spatial organisation well. (C) Single-subject ICA result from MSMSulc data shows spatial misalignment in parietal regions compared to single-subject MSMAll ICA results shown in A (from which the black outline was derived). (D) Dual regression of this subject's MSMSulc data against MSMAll group maps captures spatial organisation well, despite the observed spatial shift. These results illustrate dual regression being minimally affected by spatial misalignment. Note that the black outline in A, B, C reflects boundaries of MSMAll single subject results (shown in A), while the blue outline in D reflects boundaries of MSMSulc single subject results (shown in C). Data of *Figure 1* is available on BALSA (https://balsa.wustl.edu/study/show/0Lwm6), where all 25 components can be viewed.

DOI: https://doi.org/10.7554/eLife.44890.002

The following figure supplement is available for figure 1:

**Figure supplement 1.** The effect of spatial misalignment on dual regression estimates was tested by comparing data that were aligned using MSMAll, MSMSulc, and volumetric alignment approaches.

DOI: https://doi.org/10.7554/eLife.44890.003

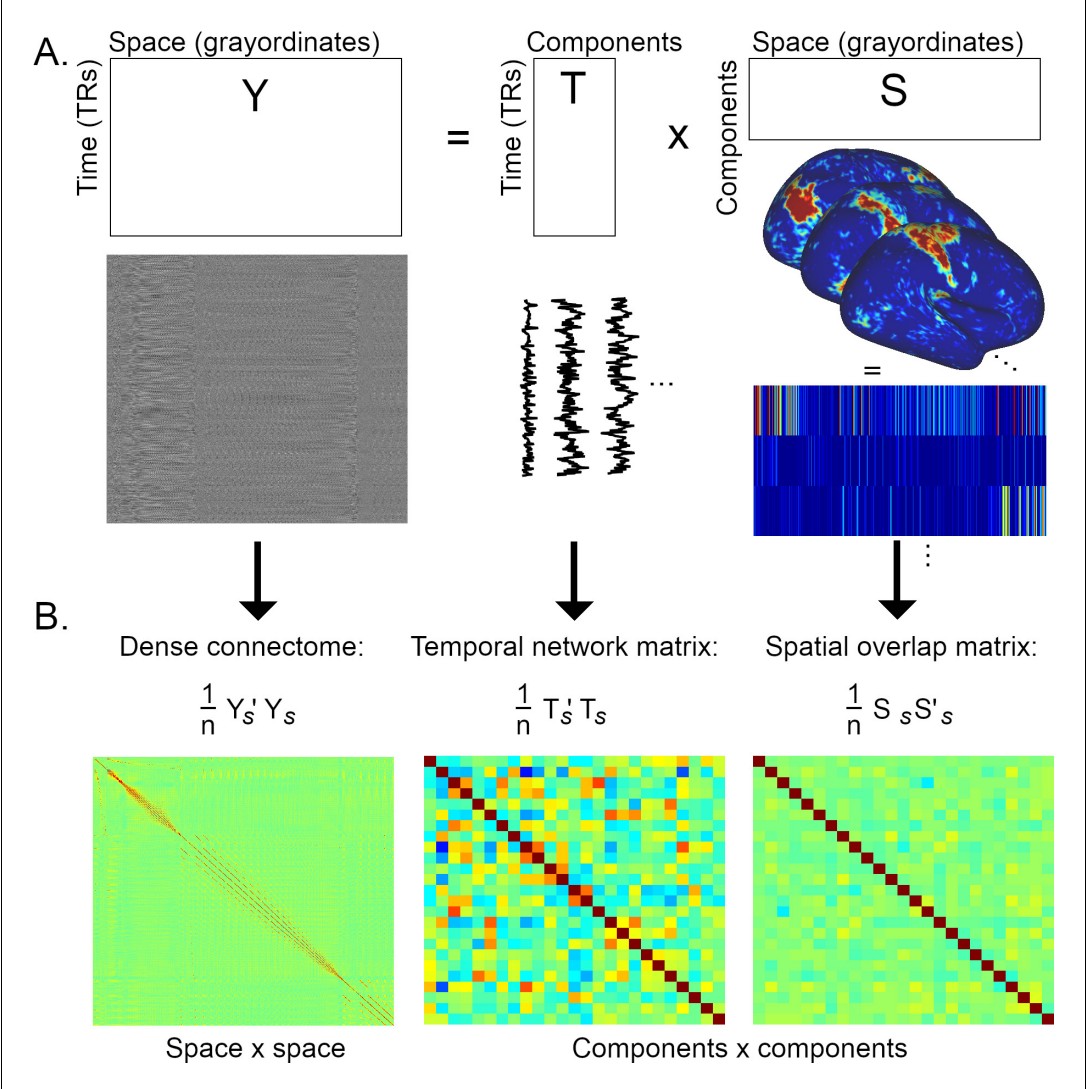

**Figure 2.** Graphical summary of the estimation of temporal network matrices and spatial overlap matrices. (**A**) Firstly, the full dataset is decomposed into a number of components/modes (the generic outer product model shown here is relevant to both ICA and PROFUMO). (**B**) Full Pearson's correlation matrices are estimated from the elements shown in A (the subscript s refers to matrices that have been standardised over time for Y and T, and standardised over space for S; n=#TRs for the dense connectome and the temporal network matrix, and n=#grayordinates for the spatial overlap matrix). The correlation matrix of the full dataset (Y) results in a voxel-to-voxel matrix referred to as the 'dense connectome'. The correlation matrices of the component maps or timeseries result in component-to-component matrices. Here, each element in the temporal network matrix describes the correlation between two timecourses (i.e., a temporal edge), whereas each element in the spatial overlap matrix describes the correlation between two spatial maps (i.e., a spatial edge).
DOI: https://doi.org/10.7554/eLife.44890.004

Group ICA was performed using all data from 1,004 HCP subjects, and the dimensionality of decomposition was fixed at 50 nodes. Following group ICA (using the temporal concatenation approach), dual regression was performed to derive subject-specific node timeseries and spatial maps. Dual regression comprises two stages, where stage one involves multiple spatial regression against group spatial ICA node maps (resulting in subject-specific node timeseries), and stage two involves multiple temporal regression against the stage one timeseries (resulting in subject-specific node spatial maps).

Subject-specific node timeseries derived from 50 nodes are correlated to estimate a 50 × 50 network matrix of temporal edges, and subject-specific node spatial maps from 50 nodes are correlated to estimate a 50 × 50 spatial overlap matrix of spatial edges. Pearson's correlation coefficient ('full

correlation') is used for both temporal and spatial edge estimates. Note that, while group spatial ICA maps are uncorrelated due to the spatial independence constraint, the subject-specific node maps derived from stage 2 of dual regression can be correlated. We subsequently directly compare the relationship between spatial overlap and temporal network matrices across all edges and subjects.

The results reveal a significant negative correlation (across edges and subjects) between ICA-DR temporal network matrix edges and ICA-DR spatial overlap matrix edges (*Figure 3*). This means that when the spatial maps for two nodes are less correlated (e.g. there is less spatial overlap), then there is more positive functional connectivity between these nodes. This negative association is surprising, because one might expect the presence of less spatial overlap between two nodes to be associated with less functional connectivity (rather than more) between those nodes.

## Temporal network and spatial overlap matrix estimation in ICA followed by dual regression

We now provide evidence that the negative association between node spatial map correlations and functional connectivity in *Figure 3* could be the result of dual regression being used on spatial ICA maps that are incorrect, due to the assumption of spatial independence being wrong (e.g. when there is spatial overlap between nodes).

Consider that neuroimaging data $Y[timepoints\ x\ voxels]$ can be summarised as a linear combination of a set of spatial maps $S[voxels\ x\ N]$ (where N = number of extracted components) and a set of timeseries $T[timepoints\ x\ N]$ following the outer product model:

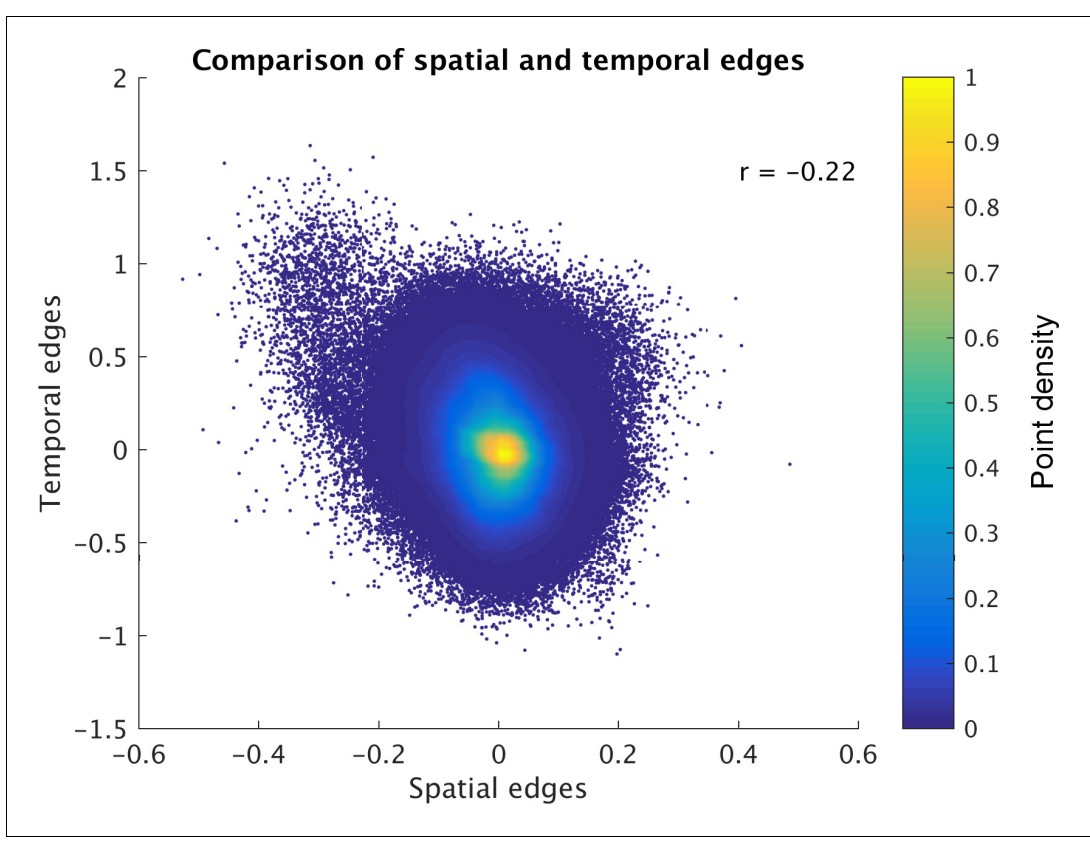

**Figure 3.** Direct comparison of spatial and temporal edges estimated with dual regression (following group ICA). Each point in the figure panels represents one edge in one subject (i.e., 1225 edges x 1004 subjects are shown). The results show that full temporal and full spatial edges obtained from ICA-DR are negatively correlated. This negative correlation largely persists when 'outlier' edges with a spatial correlation below −0.2 are removed from the comparison (r = −0.19).
DOI: https://doi.org/10.7554/eLife.44890.005

$$Y = TRS' \tag{1}$$

$$Y = TQQ'S' \tag{2}$$

Here, $R = QQ'$ $[N \, x \, N]$ is a square rotation matrix, which may be changed from the identity matrix in order to enforce independence in either the spatial or temporal domain (as is the case in ICA). The decomposition is often considered in terms of estimated component spatial maps $M$ $[N \, x \, voxels]$, and component timeseries $A$ $[timepoints \, x \, N]$.

$$M = Q'S' \tag{3}$$

$$A = TQ \tag{4}$$

If the rotation matrix $QQ'$ enforces spatial independence, $cov(M') = cov(QS)$ $[N \, x \, N]$ is equal to the identity matrix. *Equation 4* therefore shows that all between-component covariance information must be contained in $cov(A) = cov(TQ)$ $[N \, x \, N]$ in the case of spatial independence. Hence, $cov(TQ)$ as estimated from spatial ICA (or equivalently from stage 1 dual regression) reflects a weighted combination of the ground truth temporal and spatial covariance. Generally speaking, each parcellation method sits somewhere along a continuum of how the total combined temporal and spatial covariance structure is represented, as determined by the form of the rotation matrix $QQ'$. For example, both non-overlapping hard parcellations and spatial ICA represent all covariance structure temporally, whereas temporal ICA represents all covariance structure spatially (*Smith et al., 2012*). PROFUMO does not explicitly enforce orthogonality or independence in either domain, and sits along this continuum between both extremes based on model parameters and priors.

To explain the negative correlation between dual regression spatial overlap and temporal network matrices (*Figure 3*), we need to look at stage 2 of dual regression. Here, a multiple temporal regression is performed using the $A = TQ$ timeseries from *Equation (4)* to estimate node spatial maps:

$$S_{DR} = pinv(TQ) \, Y \tag{5}$$

For clarity, we provide the matrix dimensions as: $pinv(TQ)$ $[N \, x \, timepoints]$ and $Y[timepoints \, x \, voxels]$. Assuming that $TQ$ are zero mean and unit variance, the first part of the pseudo-inverse $((TQ)'TQ)^{-1}[N \, x \, N]$ is equal to the inverse of the covariance matrix (i.e., the partial correlation between stage 1 dual regression timeseries multiplied by -1). Therefore, the spatial maps obtained in *Equation (5)* ($S_{DR}$) are negatively weighted by the partial correlation between the stage 1 timeseries ($TQ$). As we saw in *Equation (4)*, $cov(TQ)$ represents a weighted combination of the ground truth spatial and temporal covariance due to the enforced spatial independence. Therefore, the inflated temporal correlations in $cov(TQ)$ will negatively weight the correlations between spatial maps, resulting in the negative correlation observed in *Figure 3*.

It is worth noting that the above theory holds in the absence of any added unstructured noise, which is expected to be present in realistic rfMRI data. Previous work has shown that thresholding can be used in the presence of noise to recover spatial overlap between group spatial ICA node maps (*Beckmann et al., 2005*). Therefore, using thresholded maps to extract timeseries may be an appropriate technique to provide more accurate measures of spatial overlap and temporal network matrices for the ICA-DR pipeline. This thresholding approach is discussed and tested in detail below in the section entitled 'Using mixture-model thresholding to improve ICA dual regression estimates'.

To show the effects of the theory described above, we performed a two dimensional simulation that includes two correlated maps (25% overlap; each map occupying 1 percent of total voxels). We generated data for 50 subjects by taking the outer product of the maps with two correlated timeseries, and entered simulated data from all 50 subjects into a spatial ICA using the temporal concatenation approach. Random noise was added to each map in each subject, but there was no systematic misalignment across simulated subject data. The full simulation was repeated across 10 instances to obtain robust results. Example group maps estimated using spatial ICA show somewhat weaker weights in the overlapping region, consistent with the independence constraint (*Figure 4A*

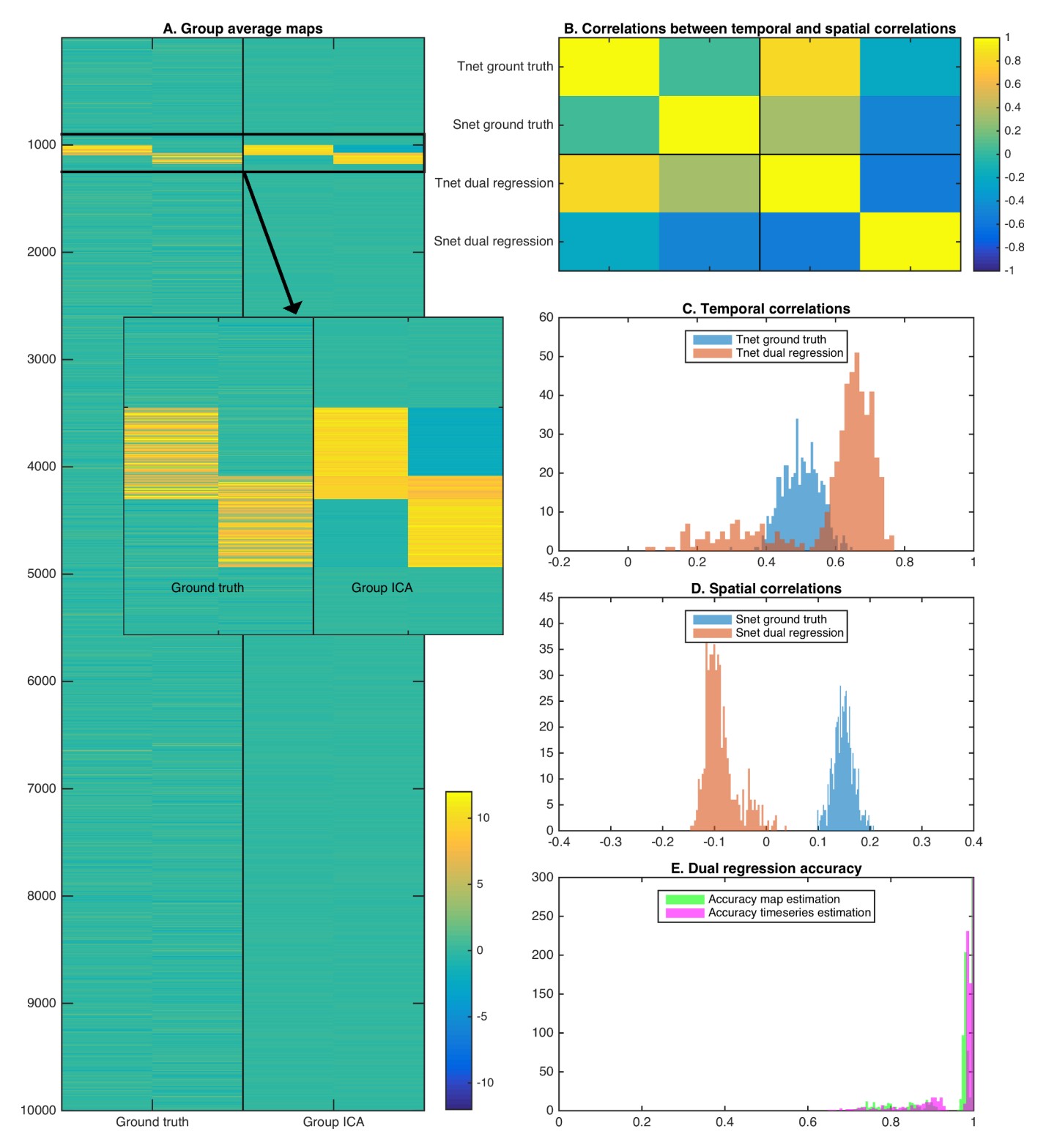

**Figure 4.** Simulation results showing the effects of assuming spatial independence when there is spatial overlap between nodes. (**A**) Spatial ICA leads to an underestimation of map weights in the overlapping area and an introduction of negative weights to meet the assumption of spatial independence. (**B**) The temporal correlation between timeseries (Tnet) as estimated by dual regression is a weighted sum of the ground truth temporal correlation and the ground truth spatial correlation, leading to a negative correlation between temporal and spatial (Snet) correlations estimated with ICA-DR. (**C**) Temporal correlations estimated with dual regression (orange) are inflated compared to the ground truth (blue). (**D**) Spatial correlations

*Figure 4 continued on next page*

*Figure 4 continued*

estimated with dual regression (orange) have a negative bias compared to the ground truth (blue). (E) Despite large shifts in temporal and spatial correlations observed in C and D, the accuracy of estimated timeseries (magenta) and spatial maps (green) is relatively high.

DOI: https://doi.org/10.7554/eLife.44890.006

and insert). The group maps are used in a dual regression pipeline to obtain estimated subject time-courses (and derived temporal correlations), and estimated subject maps (and derived spatial correlations). As described in the theory above and consistent with the results shown in real data in *Figure 3*, the derived spatial overlap and temporal network matrices are inversely correlated with each other (*Figure 4B*). As expected, the temporal network matrix edges were shifted positively compared to the ground truth (*Figure 4C*), and the spatial overlap matrix edges were shifted negatively compared to the ground truth (*Figure 4D*). Nevertheless, the individual maps and timecourses estimated with ICA-DR are highly correlated with the ground truth (*Figure 4E*). Hence, even though the first-order map and timecourse estimates from ICA-DR may have good accuracy, second-order spatial and temporal edge estimates can be more strongly affected by systematic shifts if the assumption of spatial independence is not met.

The theory and simulation above describe the effects of the presence of 'true' spatial overlap on estimated temporal and spatial correlations obtained from a traditional ICA-DR pipeline. Spatial overlap is shown to affect estimated temporal and spatial correlation such that: (i) temporal correlations between stage one dual regression timeseries are a weighted sum of ground truth spatial and temporal correlations, and (ii) spatial correlations between stage two dual regression node maps are inversely weighted by the partial correlation between timeseries.

In our previous work, we showed that simulated data containing *only* interindividual variation in node spatial maps resulted in a substantial amount of interindividual information in temporal network matrices estimated with ICA-DR (*Bijsterbosch et al., 2018*). There, the spatial information in the simulated data (i.e., the simulation 'ground truth') were subject-specific PFM maps, which are known to contain spatial overlap (*Harrison et al., 2019*; *Harrison et al., 2015*). Therefore, the theory above provides a clean (and mathematical) explanation for how the 'ground truth' spatial correlations present between PFM spatial maps can contaminate temporal correlations estimated from traditional ICA-DR.

## Evidence for the existence of overlap in real data

The theory and results in the previous sections show the potential effect of 'true' spatial overlap on results obtained from the ICA-DR pipeline. An important question is what level of overlap in the spatial organisation of large-scale brain networks is present in rfMRI data. While it is not straightforward to know the 'ground truth' functional organisation in the human brain, we present several results that can provide insights.

The first approach is to simply take subject-specific map estimates obtained from different parcellation methods (ICA-DR and PROFUMO), and to sum the grayordinate-wise weights across all maps. Subject-specific maps are first normalised (separately per node) to a maximum of 1 and thresholded such that voxel weights between −0.2 and 0.2 are set to zero to remove background noise. These steps are needed to avoid the resulting summary overlap maps being driven by either node maps with strong weights or by contributions from background weights. For each subject, a summary map is obtained by summing the absolute values at each grayordinate across all node maps. These summary maps are subsequently averaged across all subjects. The results of this simple approach reveal areas of overlap in the temporo-parietal-occipital junction (*Figure 5A*). Overlap areas are spatially consistent between ICA-DR and PROFUMO approaches, although PFM maps show somewhat more extensive regions with high overlap.

In addition, it is of interest to compare maps obtained from ICA and PROFUMO. PROFUMO does not enforce the spatial independence constraint and is therefore well suited to capture overlap. *Figure 5B* shows a direct comparison of the overlap between two matching group-level components obtained from PROFUMO and ICA. The spatial correlation between un-thresholded ICA and PFM maps is high (r = 0.84 between the red maps and r = 0.80 between the green maps). However, the spatial correlation between the two un-thresholded PFM maps is strong (r = 0.46), whereas there is

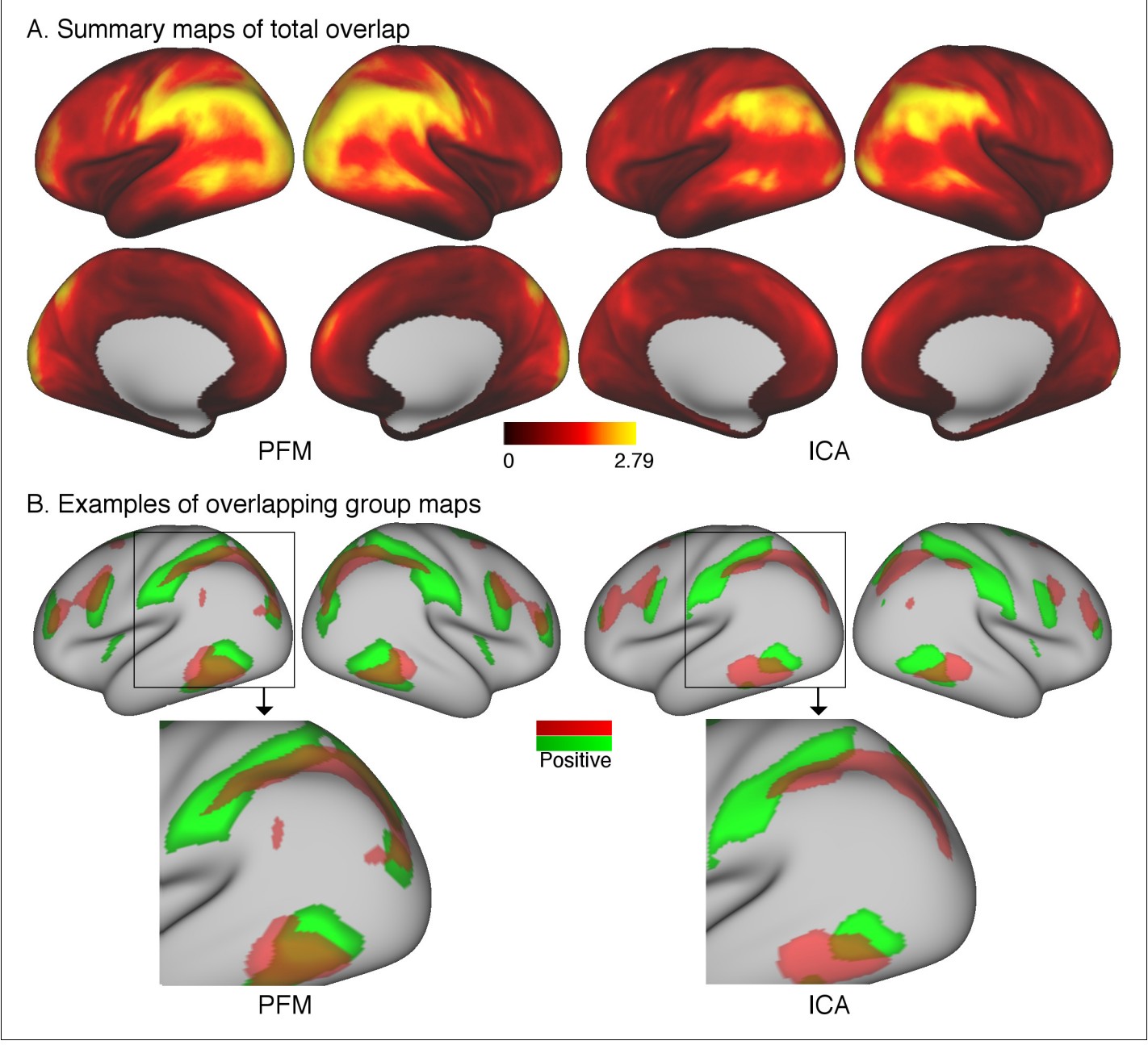

**Figure 5.** Overlap as estimated from data from the Human Connectome Project. (**A**) The absolute sum across node maps reveals a cortical pattern of overlap regions. Overlap occurs most noticeably in the temporal-parietal-occipital junction, and the spatial organisation is similar between PROFUMO results (left) and ICA-DR results (right). As expected, overlap is somewhat more extensive in PROFUMO results compared with ICA-DR. (**B**) A specific example shows that overlap is increased between PFM maps, even when highly similar spatial maps are found with ICA-DR. Data for this figure is available on BALSA (https://balsa.wustl.edu/study/show/0Lwm6) where overlap between all map pairs can be visualised.
DOI: https://doi.org/10.7554/eLife.44890.007

no correlation between the two un-thresholded ICA maps. While the thresholded maps in *Figure 5B* on the right show that ICA captures some of the overlapping regions, the extent of these is qualitatively reduced compared to PFM maps on the left.

## Using mixture-model thresholding to improve ICA dual regression estimates

We have demonstrated that if the assumption of spatial independence is incorrect, then this induces negative correlations between node spatial maps, which in turn can contaminate functional connectivity through dual regression. We now consider an alteration to traditional ICA dual regression to alleviate this contamination. The proposed method works by reducing the problematic negative spatial correlations present in the node/component spatial maps following spatial ICA. This is known as thesholded dual regression, in which a Gaussian/Gamma mixture model can be fitted to the histogram of an ICA component map to determine a threshold used to zero the background. This allows better recovery of ground truth spatial correlations in simple simulations (see Figure 3 in *Beckmann et al., 2005*).

In order to estimate timecourses such that the temporal network matrices more accurately match the ground truth, we propose to perform thresholding of the subject-specific spatial maps obtained from dual regression stage two using Gaussian/Gamma mixture modelling. Specifically, the proposed solution includes the following stages (*Figure 6*):

1. Multiple regression of subject data against group maps (identical to current stage 1)
2. Multiple regression of subject data against stage one timeseries (identical to current stage 2)
3. Thresholding of stage two spatial maps using mixture modelling (newly proposed stage 3)
4. Multiple regression of subject data against stage three thresholded maps (newly proposed stage 4)

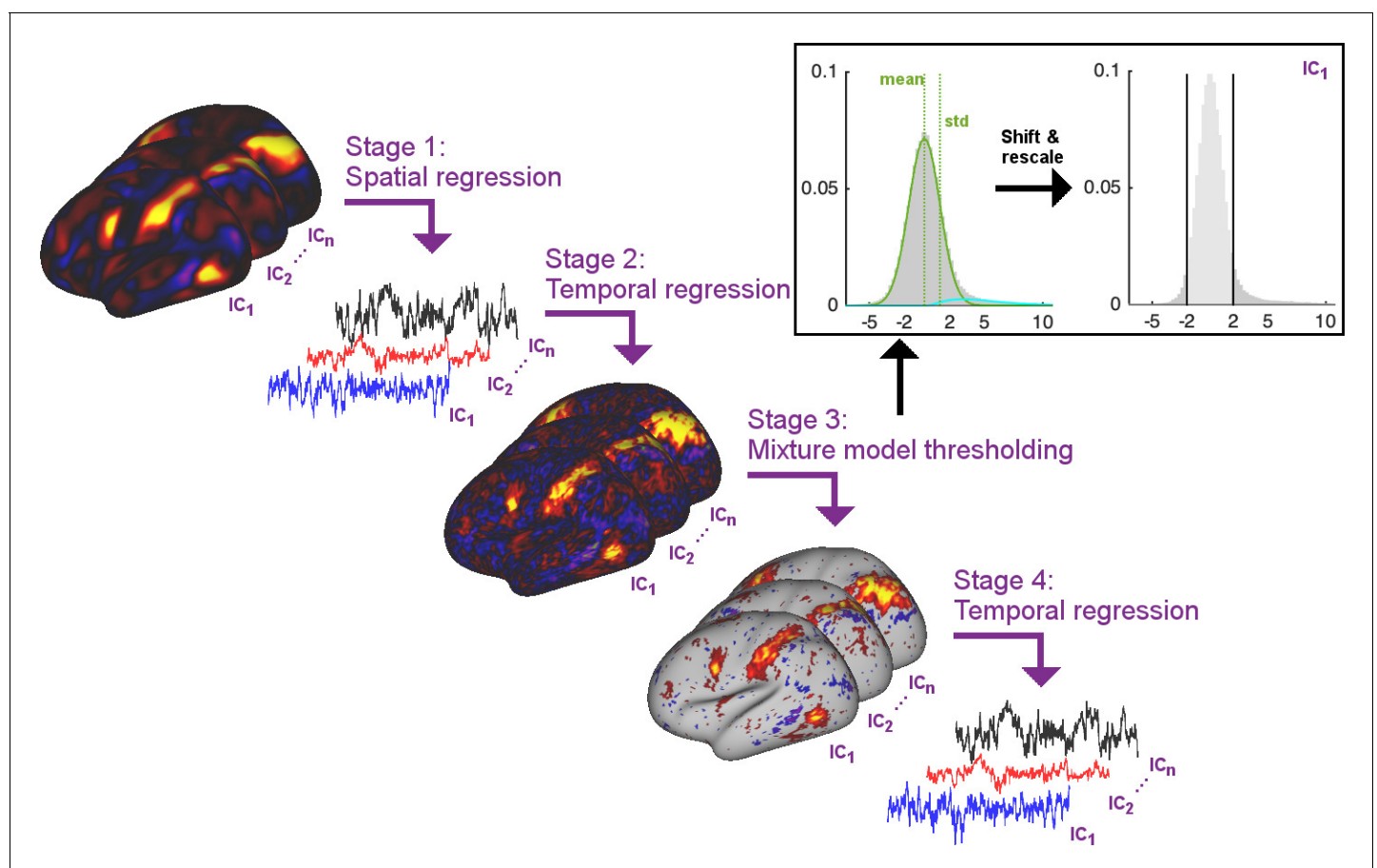

**Figure 6.** Summary schematic of the stages involved in the proposed thresholded dual regression procedure. The timeseries output from Stage four should be used to perform further node-based analyses.
DOI: https://doi.org/10.7554/eLife.44890.008

These stage four timeseries will then be used for the estimation of temporal network matrices, instead of using the intermediate timeseries from dual regression stage 1. Below, we use simulations to test this thresholded ICA-DR procedure against the standard ICA-DR pipeline and against PROFUMO.

In stage three above, mixture modelling is used to approximate the distribution of voxelwise spatial weights in a given subject-specific stage two map using a Gaussian for the background, and two Gamma distributions for positive and negative tails (*Beckmann et al., 2005*). The mean and standard deviation of the Gaussian background distribution are used to shift and rescale the map weights and a threshold of 2 is applied to set the background to zero.

Dual regression can be performed as part of two different analysis pipelines with relatively separate goals and interpretations. Firstly, dual regression can be performed in order to draw inferences regarding the spatial shape and local amplitude of networks. This analysis procedure is best performed on the un-thresholded stage two spatial maps, because subject-specific thresholding may introduce unwanted biases in the subsequent voxel-wise inference performed on the maps. Secondly, dual regression can be performed in order to extract timecourses used to estimate temporal network matrices (and hence downstream inferences based on network-matrix edge comparisons across subjects). For this second analysis pipeline we show here that using stage four timeseries instead of stage one timeseries is beneficial in order to more clearly separate spatial and temporal information (at least in the typical case of whole-brain analyses; spatial ICA performed within a small region of interest may behave differently if the number of background voxels is reduced relative to the number of 'active' voxels, such that the mixture model may not be valid). A third potential use of dual regression is to study spatial overlap matrices. While spatial relationships between networks are not typically studied, there is increasing interest in cross-subject differences in spatial network organisation. Spatial overlap matrices (which should be studied using stage three thresholded maps) reflect a relatively simple metric related to overlapping organisation and may become of increasing interest, particularly when studying associations with behaviour.

## Direct comparisons of PROFUMO and ICA accuracy using simulations

To test the accuracy of estimates obtained from PROFUMO, traditional ICA-DR, and thresholded ICA-DR (described in the previous section), we simulated an rfMRI dataset containing a ground truth set of nodes. To generate the simulations, we followed the general framework described in detail in *Harrison et al. (2019)* and *Harrison et al. (2015)*, with slight changes to some of the parameters. Briefly, the simulated data contained 10,000 voxels, 30 subjects, two runs per subject, and 600 timepoints per run. This full simulation is repeated 10 times in order to obtain well-sampled results.

In the spatial domain, a binary atlas of 100 non-overlapping contiguous parcels was generated, and random subject-specific warps were applied as a simplified representation of misalignment (with the maximum possible voxelwise displacement equal to the average parcel size). Then, 15 non-binary spatial node maps were generated by specifying weights for each atlas parcel determining how strongly it contributes to each spatial node. While the parcel identities were fixed, the strength of the weights varied from subject-to-subject. In the temporal domain, sparse and correlated 'neural' timecourses were generated and convolved with subject-specific haemodynamic response functions.

Following the generation of subject data, the full PROFUMO, traditional ICA-DR, and thresholded ICA-DR pipelines are performed to obtain group maps, subject-specific maps, and subject-specific timeseries. To test the performance of each of these approaches, we calculate (at a subject level): (i) the correlation between estimated and ground truth timeseries, (ii) the correlation between estimated and ground truth spatial maps, (iii) the correlation between estimated and ground truth results for each temporal network matrix edge (across subjects), and (iv) the correlation between estimated and ground truth results for each spatial overlap matrix edge (across subjects).

In addition, we focus on a specific subset of the edges (across all repeats of the simulation) with significantly positive ground truth spatial correlation based on a one-tailed t-test against zero (after Bonferroni correction for multiple comparison across 15*14/2 edges * 10 iterations = 1050 comparisons). A total of 126 edges (out of 1050) showed significant ground truth positive spatial correlation. The simulations are designed to have reasonable, but not excessive, amounts of spatial overlap (consistent with the results in *Figure 5A*), and therefore the spatial correlations are relatively weak, despite being significantly different from zero. We investigate these specific edges further because the tests described above are correlation-based and it is possible that interindividual variance is

captured well (leading to good results in correlation-based tests) even in the presence of a considerable bias in absolute network matrix values. We aim to test whether there is any absolute bias away from ground truth network matrix values (as measured with the positive-edge comparisons), and how well each method captures relative differences between individuals (as measured using the correlation-based tests).

The results show that the accuracy of estimated timeseries is relatively good in all of the analysis pipelines tested here (*Figure 7A*). The accuracy of spatial map estimation is superior in PROFUMO compared to both variations of the ICA-DR pipeline (*Figure 7B*). As expected, the accuracy of spatial map estimates was improved using the thresholded ICA-DR pipeline compared to the traditional ICA-DR pipeline (*Figure 7B*). Similar overall results are found for the accuracy of spatial overlap matrices (*Figure 7D*). These biases in secondary estimates for spatial and temporal edges are also found in the absence of any between-subject spatial misalignment, confirming that this effect is independent of any potential misalignment problems (*Figure 7*, *Figure 7—figure supplement 1*). When focusing on edges with significantly positive ground truth spatial correlation, PROFUMO outperformed both ICA-DR pipelines for estimating spatial edges (*Figure 7E*), whereas thresholded ICA-DR results were closest to the ground truth for estimating temporal edges (*Figure 7F*).

## Linking spatial overlap and temporal network matrices to behaviour

In our previous work we found a strong relationship between PROFUMO spatial maps and a behavioural mode that includes positive and negative traits (*Bijsterbosch et al., 2018*). Here, we repeat the canonical correlation analysis (CCA) on 1,001 HCP subjects to estimate multivariate relationships between a set of behavioural variables and a set of edges. For the edges, we test both spatial overlap and temporal network matrices obtained from one of the pipelines described above. Temporal network matrices are classically used as a proxy for neural coupling, and are commonly studied in the literature. On the other hand, spatial overlap matrices reflect spatial overlap between resting state networks, which is an aspect of functional connectivity that has not yet received attention in the existing literature. Here, we aim to test which of these different aspects of functional connectivity is most strongly and uniquely associated with individual differences in behaviour.

Details of the CCA procedure can be found in our previous work (*Bijsterbosch et al., 2018*). Briefly, CCA takes a set of behavioural measures and a set of edges as inputs and determines a linear combination of each such that the resulting scores are maximally correlated between the two sets of input variables. Following earlier work (*Bijsterbosch et al., 2018*; *Smith et al., 2015*), we use a total of 158 behavioural variables available in the Human Connectome Projects broadly covering demographic (e.g. education level), psychometric (e.g. IQ), lifestyle (e.g. substance use), and psychosocial (e.g. rule-breaking behaviour) measures. Previous work has shown a strong relationship between a behavioural population mode of variation that includes positive measures (such as IQ and self-reported life satisfaction) and negative measures (such as drug and alcohol use) (*Smith et al., 2015*).

To determine the unique variance contained in spatial overlap and temporal network matrices respectively, we adopted the same partial CCA approach that was used previously (*Bijsterbosch et al., 2018*; *Smith et al., 2015*). We regress spatial edges onto temporal edges within each method (after dimensionality reduction), and enter the residuals into a standard CCA against behavioural measures. Hence, the partial CCA results remove any shared behaviourally-relevant variance that is present in both spatial overlap and temporal network matrices within the overall method (i.e., within PROFUMO/standard ICA-DR/thresholded ICA-DR).

All of the spatial overlap and temporal network matrix results obtained with any of the three pipelines show significant associations with behaviour (*Figure 8*, blue bars). The strongest relationship with behaviour is observed for spatial overlap matrices estimated with PROFUMO ($R_{UV}$ = 0.72), which is significantly stronger than with the original ICA-DR temporal network matrix result (p=0.03), and compared with the thresholded ICA-DR temporal network matrix result (p=0.001). This finding is closely linked to our earlier work, where we reported a strong CCA result for PFM spatial maps, but there are key differences in the way brain-based inputs to the CCA were calculated between these two findings. Here, we use subject-specific spatial overlap matrices as input (i.e., only including spatial correlations between maps), whereas our previous results were based on the full set of subject spatial maps (i.e., including all spatial features in all maps).

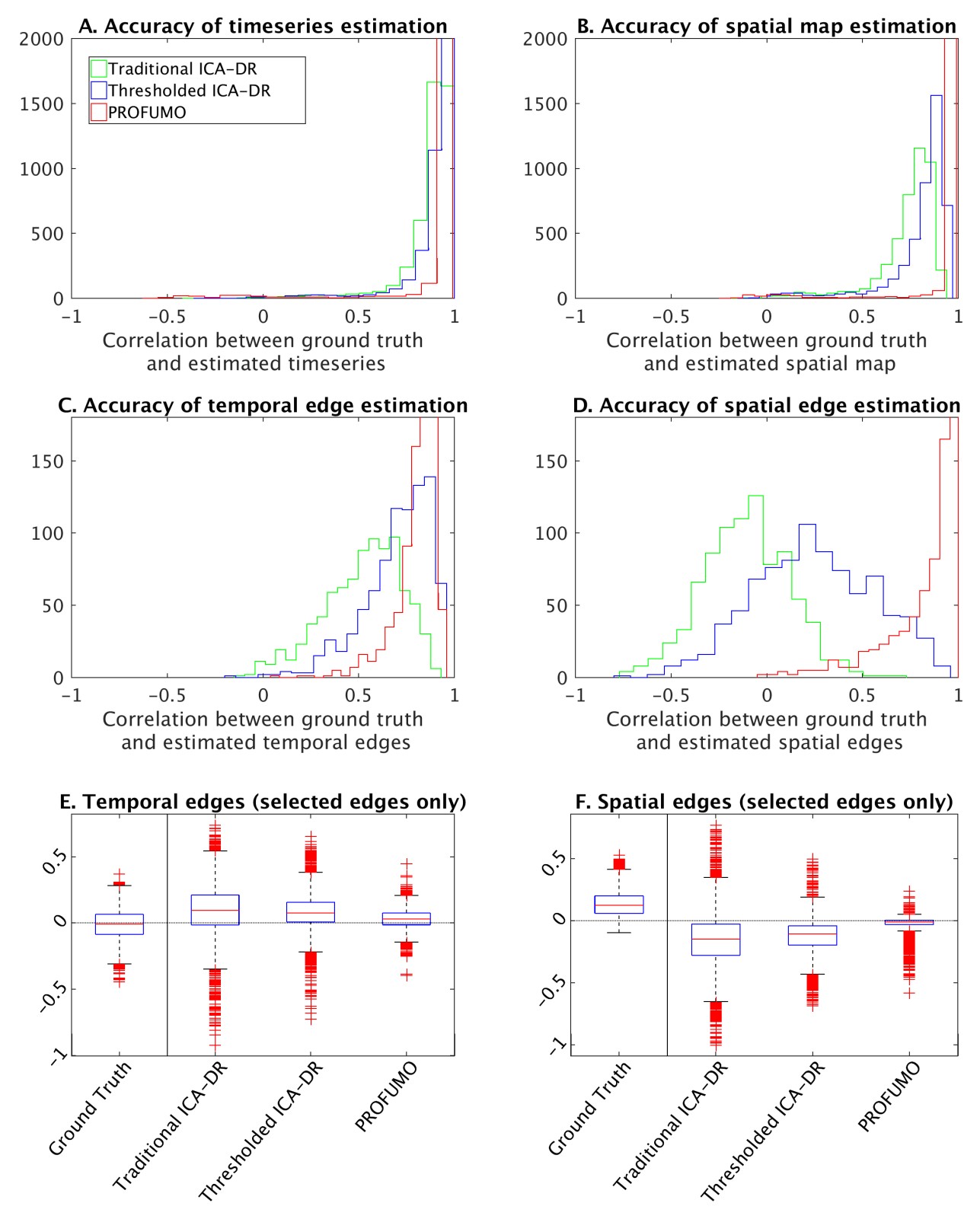

**Figure 7.** Simulation results to compare traditional ICA-DR, thresholded ICA-DR, and PROFUMO performance. (**A**) Correlations between ground truth and estimated subject timeseries is high across all three methods. (**B**) Correlations between ground truth and estimated subject spatial maps is highest in PROFUMO results (red), and improved in thresholded ICA-DR results (blue) compared to traditional ICA-DR results (green). Similar results are found for cross-subject correlations of temporal network matrix edges (**C**), and for cross-subject correlations of spatial overlap matrix edges (**D**). Results in A-D

*Figure 7 continued on next page*

*Figure 7 continued*

are shown for all maps that achieved a minimum group-average spatial correlation between ground truth and estimated maps of r = 0.5 across the three methods. Figures E and F show results for a subset of edges with significantly positive spatial correlation. Here, the first 'ground truth' column shows the distribution of ground truth edge strengths, whereas columns 2–4 show the difference between estimated and ground truth edge strengths (i.e., results in columns 2–4 that are closest to zero are best).

DOI: https://doi.org/10.7554/eLife.44890.009

The following figure supplement is available for figure 7:

**Figure supplement 1.** Comparison across different simulations for the accuracy of temporal edge estimation (A), and for the accuracy of spatial edge estimation (B).

DOI: https://doi.org/10.7554/eLife.44890.010

To facilitate the interpretation of these findings in light of our earlier work (*Bijsterbosch et al., 2018*), we include a movie of spatial overlap as a function of the most significant behavioural CCA mode (*Figure 8—video 1*). Here, we followed the same procedure described previously to generate interpolated movie frames for each of the 50 PFM maps, and subsequently summed the grayordinate-wise weights across all 50 maps following normalisation (as for *Figure 5A*). The resulting movie (*Figure 8—video 1*) shows that spatial overlap systematically varies along with the network-specific spatial variation that we showed previously (*Bijsterbosch et al., 2018*). Hence, these results show that spatial overlap (and resulting correlation) is a key behaviourally-relevant aspect of spatial information.

Out of the results that reach significance in *Figure 8*, subject behavioural weights are significantly correlated with the previously reported positive-negative population mode of behaviour (*Bijsterbosch et al., 2018*; *Smith et al., 2015*) for all results except for PFM temporal netmats. The PFM temporal netmats are instead linked to variables such as blood pressure (diastolic and systolic), hematocrit values, and alcohol use, therefore representing a more physiological population mode. Correlation with the positive-negative mode was reduced for all partial CCA results compared with full CCA results ($R_{full}$ = 0.14 to $R_{partial}$ = 0.01 for ICA-DR thresholded temporal network matrices; $R_{full}$ = 0.42 to $R_{partial}$ = 0.21 for ICA-DR original temporal network matrices; $R_{full}$ = 0.39 to

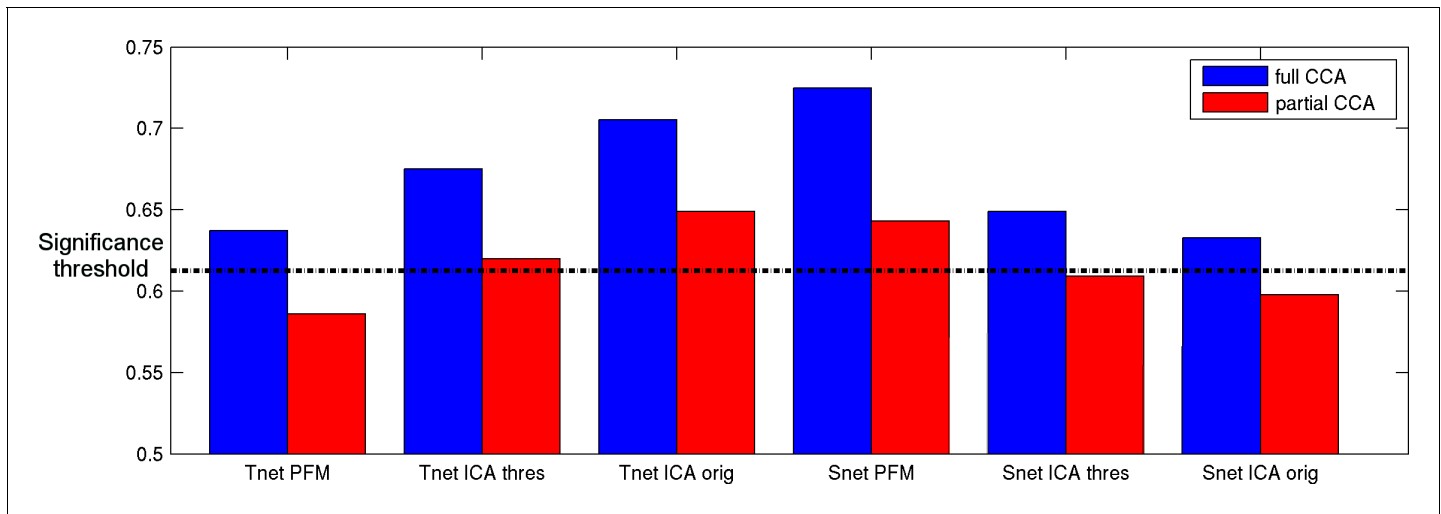

**Figure 8.** Full and partial CCA results between spatial overlap (Snet) or temporal network (Tnet) matrices against behaviour. The strongest association with behaviour is found for PFM spatial edges. Temporal network matrices estimated with thresholded ICA-DR (thres) are less strongly linked to behaviour than those estimated with traditional ICA-DR (orig), and show a greater reduction when partialling out matching spatial edges. The dashed line indicates the mean p<0.05 significance level from permutation testing.

DOI: https://doi.org/10.7554/eLife.44890.011

The following video is available for figure 8:

**Figure 8—video 1.** To facilitate direct comparison with the results presented in our earlier work (*Bijsterbosch et al., 2018*), we created movies of spatial overlap along the axis of the behavioural CCA mode of population covariation.

DOI: https://doi.org/10.7554/eLife.44890.012

$R_{partial}$ = 0.16 for PFM spatial overlap matrices). Hence, the partial results were less similar to the positive-negative behavioural mode than the full CCA results, indicative of shared behaviourally-relevant information between spatial overlap and temporal network matrices in all of the approaches that were tested. Specifically, the amount of overall shared variance between temporal network and spatial overlap matrices was 29.6% for original ICA-DR, 26.8% for thresholded ICA-DR, and 24.5% for PROFUMO. Hence, these results show reduced conflation of temporal and spatial information in both thresholded ICA-DR and PROFUMO compared with traditional ICA-DR. Interestingly, the partial CCA result for the ICA-DR thresholded temporal network matrix was no longer significantly linked to the positive-negative behavioural mode ($R_{partial}$ = 0.01, $p_{Bonferroni}$ = 1), supporting the interpretation that cross-subject behavioural variability in temporal network matrices is largely driven by spatial information.

## Discussion

We previously showed that the spatial topographical organisation of functional networks is most strongly predictive of cross-subject variability in behaviour, and that functional connectivity (temporal) network matrices contain little unique trait-level cross-subject information that is not also reflected in spatial maps (*Bijsterbosch et al., 2018*). Here, we aimed to determine and address the reason for this conflation of temporal and spatial sources of interindividual variability, focussing on the pipeline that uses ICA for parcellation and dual regression for the estimation of subject maps and timeseries. Our results reveal that functional networks are often spatially overlapping (*Figure 5*), and that individual differences in the amount of spatial overlap (and resulting spatial correlation) is likely to explain our previous results. These findings support the main results from our earlier article that estimated temporal correlations are (in part) driven by spatial information (*Bijsterbosch et al., 2018*). However, our new findings show that these results should not be interpreted as spatial misalignment (as we suggested in our earlier work), but are rather the result of spatially overlapping functional organization. Spatial overlap is underestimated in spatial ICA (due to the statistical independence constraint; *Figure 4A*), systematically affecting downstream dual regression steps, such that the resulting temporal network matrices are negatively weighted by any spatial correlations present in the underlying true networks. These effects, discussed mathematically in section 'Temporal network and spatial overlap matrix estimation in ICA followed by dual regression', and shown in simulated (*Figure 4*) and real data (*Figure 3*), directly explain our previous dual regression-based results. Importantly, these methodological aspects are highly relevant for research into brain-behaviour relationships for two reasons. Firstly, they influence the interpretation of existing research using temporal network matrices (which likely reflect a combination of spatial and temporal information). Secondly, our findings suggest that patterns of spatial overlap may be highly sensitive to individual differences in behaviour and warrant further research.

As discussed in the introduction, one potential reason for our earlier findings could have been linked to residual spatial misalignment between group and subject maps (*Bozek et al., 2018*; *Demirtas et al., 2018*; *Kong et al., 2019*). The results in *Figure 1* reveal that dual regression obtains relatively accurate estimates in the presence of minor misalignment (as modelled with MSMAll compared with MSMSulc), suggesting that spatial misalignment may not contribute strongly when care is taken to adopt surface-based alignment methods and subject-specific node-definition approaches. Therefore, our earlier findings (in HCP data) are more likely explained by the biases introduced as a result of spatial overlap, and not by residual spatial misalignment. However, dual regression was not able to account for more extensive misalignment which is seen when using volume-based alignment procedures (*Coalson et al., 2018*). Therefore, it is likely that misalignment may influence functional connectivity estimates for analyses performed in volumetric space (*Smith et al., 2011*). Similarly, misalignment is likely to play a larger role when using anatomically-derived hard parcellations that are unlikely to match the subject-specific functional organisation well.

In this work we have broadly considered spatial edges as a distinct form of functional connectivity (to be differentiated from temporal edges; see *Figure 2*). However, the correct interpretation of spatial edges is an important new research question that requires further investigation. On the level of neural circuitry, neural organisations that may give rise to observed spatial overlap in fMRI include: (i) independent co-located (at fMRI resolution) neural populations, (ii) fast dynamic switching between network membership, (iii) multiplicity of hierarchically organised functional representations,

(iv) concurrent contributions to multiple networks (i.e., integration). Determining the neural basis of spatial overlap will inform the correct interpretation of spatial edges by addressing questions such as: 'is spatial overlap indicative of coupling?", and 'do spatial edges fall within the definition of functional connectivity?".

The detailed explanation of our previous findings presented here only applies to the results that were obtained using a dual regression pipeline. Mathematical differences in timeseries estimation imply that the theory described here cannot explain our earlier findings obtained using a masking procedure with hard parcellations (i.e., Yeo and HCP_MMP1.0) (*Glasser et al., 2016*; *Yeo et al., 2011*). We are currently undertaking similarly detailed investigations to further understand our previous results in these hard parcellations. Early qualitative observations suggest the presence of complex patterns of partial overlap between multiple extended networks. This complex overlap may not easily be modelled by increasing the dimensionality to allow separation of previously overlapping voxels into one new parcel, and non-overlapping voxels into other parcel(s) (which would allow results to be unbiased, particularly at a sufficiently high dimensionality). As such, our preliminary results suggest that hard parcellations are affected by overlapping functional organisation due to the mis-estimation of spatial nodes and resulting mixing of extracted timeseries (*Smith et al., 2011*). Further work is needed to test this hypothesis and to develop parcellation methods that address this issue.

The biases described in this work can be addressed using a relatively straightforward extension to the existing dual regression pipeline. Specifically, we suggest a further regression step after standard dual regression, that uses thresholded subject maps to estimate timeseries. It has previously been shown that thresholded maps (using Gamma-Gaussian mixture modelling) better capture spatial overlap (*Beckmann et al., 2005*; *Bielczyk et al., 2018*). Here, we explicitly test this approach and report greater accuracy in the estimation of timeseries, spatial maps, and both temporal and spatial overlap matrices when compared with traditional dual regression (*Figure 7*). When dual regression is performed with the aim of extracting timeseries and estimating temporal network matrices (as opposed to statistical tests of spatial map shape and amplitude), we recommend adopting this thresholded dual regression extension in order to accurately estimate node timeseries. While this approach may lead to somewhat lower associations with behaviour (*Figure 8*), it is driven more purely by uniquely temporal connectivity information, rather than conflating both temporal and spatial connectivity. Thresholded dual regression will be implemented in the next version of FSL dual_regression (https://fsl.fmrib.ox.ac.uk/fsl/fslwiki/DualRegression).

Our results show that spatial correlations obtained from PROFUMO are most strongly linked to behaviour (*Figure 8*). This finding is an indirect replication of our earlier work (*Bijsterbosch et al., 2018*), but here it is driven purely by correlations between spatial maps, while we previously entered the cross-subject covariance matrix of the full weighted spatial maps. Hence, this result indicates that cross-subject variance in spatial correlations (rather than in more complex aspects of spatial organisation) are behaviourally-relevant. Unfortunately, spatial correlations obtained from thresholded dual regression are not as strongly associated with behaviour (*Figure 8*), which is likely linked to the fact that PROFUMO outperforms thresholded dual regression in the accuracy of estimated spatial maps and correlations (*Figure 7*).

The methodological challenge of accurately modelling spatial overlap in network organisation that is addressed here has important implications for a number of key neuroscientific questions. Our findings show that spatial overlap was prominent in the temporo-parietal-occipital junction and within higher visual areas along the lateral wall (*Figure 5*). The hierarchical functional organisation of V1 is relatively well-understood to represent overlapping receptive fields with selectivity for stimulus orientation, length, width, or colour (*Van Essen and Maunsell, 1983*). On the other hand, the complex functional organisation in the inferior parietal lobule, and its involvement in wide-ranging perceptual and cognitive functions, is the topic of ongoing research and debate (*Carter and Huettel, 2013*; *Igelström and Graziano, 2017*; *Lin et al., 2018*; *Mars et al., 2012*). Further evidence of the highly complex spatial nature of functional organisation can also be found in recent gradient-based analyses of cortical topography (*Haak et al., 2018*; *Margulies et al., 2016*; *Marquand et al., 2017*). These spatial complexities are largely ignored in most current connectomics approaches, where the use of group-based anatomical hard atlases (such as the AAL) is still highly pervasive. This likely limits the ability to gain further insights into the organisation and function of cortical regions such as the temporo-parietal-occipital junction. As such, the results presented here and elsewhere emphasise

the need for modern connectomics research to acknowledge and appropriately account for the presence of highly complex, hierarchically overlapping functional organisation.

It is possible that the overlapping organisation identified here may be linked to complex patterns of inter-digitation between large-scale networks (*Braga and Buckner, 2017*). We did not find any evidence for inter-digitated patterns here, although the results in *Figure 5* (and similar observations at the individual subject level) do show adjacency of two separate networks in a number of distinct cortical zones, consistent with the findings of Braga et al. A larger number of data points per subject may be needed to estimate further detailed patterns of inter-digitation in regions that we refer to here as 'overlapping' (e.g. there is a roughly three-fold increase in the Braga et al dataset compared to HCP data). Alternatively, a truly overlapping hierarchical spatial organisation may be represented as inter-digitation in seed-based correlation results (as opposed to multivariate decomposition methods), because correlations may effectively be reduced in regions of overlap relative to neighbouring non-overlapping regions, creating inter-digitated patterns. A further possibility is that individual differences in cortical folding associated with brain volume may influence both observed spatial overlap and inter-digitation (*Toro et al., 2008*). Indeed, we previously showed a strong brain-behaviour relationship for fractional surface area (Supplementary file 1B in *Bijsterbosch et al., 2018*), and recent work has reported broad associations between structural brain measures and behaviour (*Arenas et al., 2018*). Hence, further work is needed to understand the relationship between structural morphology and overlapping or inter-digitated functional organization.

In conclusion, we replicate our previous work showing that spatial topographical network organisation is most strongly linked to behaviour (*Figure 8*), and additionally show that cross-subject variability in spatial overlap between complex cortical networks is a key source of behaviourally-relevant information. Furthermore, we show that this spatially overlapping network structure is underestimated when using a common parcellation technique (ICA), resulting in the conflation of temporal and spatial connectivity information in derived temporal network matrices (from dual regression). We present a solution that obtains more accurate estimates of temporal and spatial overlap matrices based on thresholded spatial maps.

## Materials and methods

### Dataset
Data from the Human Connectome Project S1200 release were used including 1004 subjects with 4800 resting state timepoints (*Van Essen et al., 2013*). The data were preprocessed using the HCP minimally preprocessing pipeline (*Glasser et al., 2013*), followed by FIX cleanup of artificial components obtained from single-run ICA (*Smith et al., 2013*). Three subjects were excluded for CCA purposes due to incomplete genetic information (i.e., N = 1001 for CCA).

### Simulation 1
The simulation for Figure 4 contained two-dimensional spatial maps with 10,000 voxels. A total of 50 subjects with 200 timepoints each were simulated, and the full simulation was repeated 10 times. Two spatial nodes were generated using a Laplacian distribution ($\mu = 0$; $\sigma = 0.5$) for the background and linearly added uniformly distributed (ranging between 2-12) weights. Each spatial node included 100 voxels (out of 10,000), with 25% overlap between the two spatial nodes to ensure ground truth spatial correlation. Normally distributed timeseries were generated for each node, and a shared normally distributed timeseries was added to both to ensure ground truth temporal correlation. Simulated data for each subject was generated by taking the outer product between simulated timeseries and spatial maps, resulting in a 10,000 x 200 dataset. For each repeat, these datasets were concatenated across all 50 subjects before running group ICA, following by dual regression.

### Simulation 2
A second simulation was performed to enable a direct comparison between the results from original ICA-DR, thresholded ICA-DR, and PROFUMO. The key differences between simulation 1 and simulation two are: (a) a spatial model that builds complex modes from an atlas of contiguous parcels, and (b) a temporal model of the 'neural' timeseries that is convolved with a hemodynamic response

function. Simulation two was based on previous work (*Harrison et al., 2019*; *Harrison et al., 2015*), and is described in more detail below:

### Atlas generation

The 10,000 voxels were split into 100 contiguous parcels, with the parcel widths drawn from a Dirichlet distribution. The parcel weights were then drawn from a Gamma distribution. The gradient of the warp field was generated by convolving random Gaussian noise with a boxcar function, passing through a nonlinearity to limit the range to [−1,1] (i.e., to ensure the warp remained invertible).

### Node generation

Each node was formed from a number of spatially contiguous regions. The number of regions followed a Poisson distribution, the total number of parcels followed a beta distribution, and the number of parcels per region followed a Dirichlet distribution. Several regions were made to be anticorrelated (i.e., were given negative weights). The parcel weights again followed a gamma distribution.

### Time course generation

'Neural' time courses were simulated at 0.1 Hz. The frequency spectra of these were randomly generated, with a bias towards low frequencies. Correlations were induced based on group, subject, and run covariance matrices each drawn from a Wishart distribution. Finally the time courses were sparsified by setting sub-threshold time points to zero.

### HRF convolution

The time courses were convolved with random draws from the FLOBS basis set, with a unique HRF being generated for every subject.

### Outer product model

The time courses and spatial maps were combined by taking the outer product, and a weak nonlinearity was applied to the resulting voxelwise timecourses to simulate saturation of the HRF in regions exhibiting high levels of activity.

### Noise model

Finally, noise was added to the BOLD signal. This consisted of a structured and unstructured noise subspace: the structured subspace consisted of a set of 'confounds', which consisted of the outer product of Gaussian spatial maps and time courses. The unstructured noise was weakly non-Gaussian, following a Student's t-distribution.

The code, containing all used parameter values, is available from https://git.fmrib.ox.ac.uk/samh/PFM_Simulations (*Harrison, 2019*; copy archived at https://github.com/elifesciences-publications/PFM_Simulations).

## Mixture modelling

Spatial maps for resting state fMRI networks are commonly relatively sparse, with a large proportion of voxels or grayordinates considered to be part of the background (i.e., many voxels have relatively low weights and do not contribute to the network). In the presence of additive Gaussian noise, we can therefore model the distribution of spatial weights in any single ICA components using a mixture of one Gaussian distribution (for the background) and two Gamma distributions (for the positive and negative aspects of the ICA networks). Subsequently, the mean and standard deviation of the Gaussian distribution (background) are used to shift and rescale the distribution of spatial weights for each map, and a threshold of ±2 is used to threshold the spatial map such that background voxels are set to zero. Previous work has shown that this threshold procedure can accurately capture ground truth spatial correlations (*Beckmann et al., 2005*). Here, we adopt mixture model thresholding in a proposed extension to dual regression designed to estimate ICA timeseries and derived correlations with improved accuracy.

## Data and code availability
HCP data are distributed from the Connectome Coordination Facility (https://www.humanconnectome.org/). The simulation code is available from https://git.fmrib.ox.ac.uk/samh/PFM_Simulations. Brain data for *Figures 1* and *5* are available on Balsa (https://balsa.wustl.edu/study/show/0Lwm6).

## Acknowledgements

Data were provided by the Human Connectome Project, WU-Minn Consortium (Principal Investigators: David Van Essen and Kamil Ugurbil; 1U54MH091657) funded by the 16 NIH Institutes and Centers that support the NIH Blueprint for Neuroscience Research; and by the McDonnell Center for Systems Neuroscience at Washington University. CFB acknowledges support from The Netherlands Organization for Scientific Research (NWO, grant no 864.12.003). We are grateful for funding from the Wellcome Trust (grants 098369/Z/12/Z and 091509/Z/10/Z). The Wellcome Centre for Integrative Neuroimaging is supported by core funding from the Wellcome Trust (203139/Z/16/Z).

## Additional information

### Funding

| Funder | Grant reference number | Author |
| --- | --- | --- |
| Nederlandse Organisatie voor Wetenschappelijk Onderzoek | 864.12.003 | Christian F Beckmann |
| Wellcome | 098369/Z/12/Z | Stephen M Smith |
| Wellcome | 091509/Z/10/Z | Stephen M Smith |
| Wellcome | 203139/Z/16/Z | Stephen M Smith |

The funders had no role in study design, data collection and interpretation, or the decision to submit the work for publication.

### Author contributions
Janine Diane Bijsterbosch, Conceptualization, Formal analysis, Investigation, Methodology, Writing—original draft; Christian F Beckmann, Methodology, Writing—review and editing; Mark W Woolrich, Software, Writing—review and editing; Stephen M Smith, Conceptualization, Supervision, Writing—review and editing; Samuel J Harrison, Conceptualization, Software, Formal analysis, Writing—review and editing

### Author ORCIDs
Janine Diane Bijsterbosch  https://orcid.org/0000-0002-1385-9178
Samuel J Harrison  https://orcid.org/0000-0002-5886-2389

### Ethics
Human subjects: HCP data were acquired using protocols approved by the Washington University institutional review board (Mapping the Human Connectome: Structure, Function, and Heritability; IRB # 201204036). Informed consent was obtained from subjects. Anonymised data are publicly available from ConnectomeDB (db.humanconnectome.org; Hodge et al., 2016). Certain parts of the dataset used in this study, such as the age of the subjects, are available subject to restricted data usage terms, requiring researchers to ensure that the anonymity of subjects is protected (Van Essen et al., 2013).

### Decision letter and Author response
Decision letter https://doi.org/10.7554/eLife.44890.020
Author response https://doi.org/10.7554/eLife.44890.021

# Additional files

## Supplementary files

• Transparent reporting form

DOI: https://doi.org/10.7554/eLife.44890.013

## Data availability

Simulation analysis scripts are available on git (https://git.fmrib.ox.ac.uk/samh/PFM_Simulations; copy archived at https://github.com/elifesciences-publications/PFM_Simulations). Source data files for Figures 1 and 5 will be made available on BALSA: https://balsa.wustl.edu/study/show/0Lwm6.

The following dataset was generated:

| Author(s) | Year | Dataset title | Dataset URL | Database and Identifier |
|---|---|---|---|---|
| Janine Diane Bijsterbosch, Christian F Beckmann, Mark W Woolrich, Stephen M Smith, Samuel J Harrison | 2019 | Study: The relationship between spatial configuration and functional connectivity of brain regions revisited | https://balsa.wustl.edu/study/show/0Lwm6 | BALSA, 0Lwm6 |

The following previously published dataset was used:

| Author(s) | Year | Dataset title | Dataset URL | Database and Identifier |
|---|---|---|---|---|
| Van Essen D, Ugurbil K | 2017 | Human Connectome Project: WU-Minn HCP consortium | https://www.humanconnectome.org/study/hcp-young-adult | Young Adult, S1200 |

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
