## [Decision Letter]

Thank you for submitting your article "The relationship between spatial configuration and functional connectivity of brain regions revisited" for consideration by *eLife* as a Research Advance. Your article has been reviewed by Daniel S Margulies (Reviewer #1) and Jakob Seidlitz (Reviewer #2), and the evaluation has been overseen by Chris Honey, the Reviewing Editor, and Richard Ivry as the Senior Editor.

The reviewers have discussed the reviews with one another and the Reviewing Editor has drafted this decision to help you prepare a revised submission.

Summary

This manuscript by Bijsterbosch and colleagues extends their prior work describing the impact of spatial confounds on the variation in functional connectivity across individuals, and its connection to variation in behavior. Here, the authors test several specific hypotheses to account for their prior findings. They do this by examining two axes of potential bias: i) algorithms to derive spatial and temporal functional connectivity domains, and ii) methods for generating "soft" (data-driven) and "hard" (a priori) parcellations. Furthermore, they provide both empirical evidence and quantitative support via simulations in support of these investigations. Ultimately, the authors conclude that inter-individual behavioral variability is best explained by the spatial overlap between networks, which affects the extraction of soft parcellations. The manuscript is clearly written, and the analyses and results are presented in a manner that makes the logic of the study straightforward.

Overall, the reviewers were impressed by the rigor and logical progression of the manuscript, and the effort to include results from different surface alignment algorithms as well as a comparison with volumetric registration. The reviewers also appreciated the discussion and comparison of dual regression methodologies versus PROFUMO, and the clear differences in assumptions that are made with each.

Essential revisions

1A) The maps of network overlap, presented in Figure 4A, appear spatially consistent with prior findings of the relative degree of cortical folding (Toro et al., 2008). As cortical morphology may impact on functional connectivity (for example, differences between adjacent sulci and gyri), is it possible that the network overlap results are driven in part by cortical morphology? This issue is distinct from the point raised in the discussion regarding network interdigitation, but the impact of morphology on the results could be the same. A post hoc test of whether measures of cortical surface area and volume account for the brain-behavior relationship observed in the CCA analysis might be sufficient to assess whether this concern is relevant.

1B) Another possible interpretation of the topography in Figure 4: the areas of most overlap appear to be in areas that have been shown to be prone to problems in cortical reconstruction (mostly due to excess head motion). In light of the lack of influence of surface registration method (as discussed above and in the manuscript), can the authors speculate/discuss this pattern of overlap?

2) The Discussion mentions 'primary visual cortex' (V1) as being an area of high overlap (Figure 4). However the regions appear to be located predominantly within higher visual areas along the lateral wall (rather than within the calcarine sulcus). Please justify or correct this claim.

3) As the core brain-behavior finding from the CCA analysis is from Snet PFM, would there be any way to visualize the pattern of spatial overlap associated with the components (along the lines of the figures shown in Figure 4A)? This might help in interpreting the results in the context of prior findings.

4) The Discussion mentions that ongoing work explores why the brain-behavior relationships are also observed when hard parcellations are used (e.g. the Yeo et al. and Glasser et al. parcellations). The authors speculated that: "it is possible that [hard] parcellation methods are therefore unable to isolate overlap into a distinct parcel (which would allow results to be unbiased, particularly at a sufficiently high dimensionality), and instead parcel boundaries in regions of overlap are determined by the network with the strongest amplitude (or lowest cross-subject variance), leading to mixing of extracted timeseries". Are the authors now able to confirm this speculation? More generally, can the authors explain why the brain-behavior variance explained is so similar when using hard parcellations and when using dual regression approaches (Figure 1A of Bijsterbosch et al., 2018)? Should this be the case if the dual regression methods are subject to these spatial overlap / spatial independence effects, while hard parcellations are not?

---

## [Author Response]

Essential revisions1A) The maps of network overlap, presented in Figure 4A, appear spatially consistent with prior findings of the relative degree of cortical folding (Toro et al., 2008). As cortical morphology may impact on functional connectivity (for example, differences between adjacent sulci and gyri), is it possible that the network overlap results are driven in part by cortical morphology? This issue is distinct from the point raised in the discussion regarding network interdigitation, but the impact of morphology on the results could be the same. A post hoc test of whether measures of cortical surface area and volume account for the brain-behavior relationship observed in the CCA analysis might be sufficient to assess whether this concern is relevant.

We agree with the suggestion that underlying cortical morphometry may impact on functional connectivity, and could potentially underlie some of the results reported here. We previously assessed the brain-behavior relationship for cortical surface area using CCA as part our earlier paper (Supplementary File 1B: https://doi.org/10.7554/eLife.32992.028). The findings showed that there is indeed a significant brain-behaviour relationship for Fractional Surface Area (based on the HCP_MMP1.0 parcellation), which is further supported by recent preprint findings showing brain-behaviour associations for multiple structural measures (Arenas, et al., 2018). Further research will be needed to directly relate morphometry and functional organisation in order to establish the relationship between functional overlap and cortical folding. These considerations are included in the Discussion:

“A further possibility is that individual differences in cortical folding associated with brain volume may influence both observed spatial overlap and inter-digitation (Toro et al., 2008). Indeed, we previously showed a strong brain-behaviour relationship for fractional surface area (Supplementary File 1B in (Bijsterbosch et al., 2018)), and recent work has reported broad associations between structural brain measures and behaviour (Arenas et al., 2018). Hence, further work is needed to understand the relationship between structural morphology and overlapping or inter-digitated functional organization.”

1B) Another possible interpretation of the topography in Figure 4: the areas of most overlap appear to be in areas that have been shown to be prone to problems in cortical reconstruction (mostly due to excess head motion). In light of the lack of influence of surface registration method (as discussed above and in the manuscript), can the authors speculate/discuss this pattern of overlap?

As the reviewer points out, research has shown relatively widespread correlations between cortical thickness estimates from Freesurfer and head motion (Reuter et al., 2015), including in some of the overlap areas reported in our manuscript. However, these effects are highly unlikely to play a role in the findings we report here. For HCP data, the quality of the structural data and of the Freesurfer pipelines is tightly controlled and checked manually for all subjects, and poor quality structural scans were flagged and repeated to achieve optimal data quality (Glasser et al., 2013; Marcus et al., 2013), leading to very high quality surface projections. The MSMAll alignment procedure incorporates functional information in addition to surface folding patterns, and is therefore expected to be minimally affected by errors in cortical projection. While there are substantial improvements in MSMAll alignment compared to MSMSulc alignment (Coalson, Van Essen, and Glasser, 2018), we observe highly similar CCA results against behaviour regardless of the alignment procedure and we show here that dual regression performs robustly in both types of surface-based alignment. These findings are in line with the state-of-the-art quality of HCP surface projections, suggesting that cortical reconstruction errors do not contribute to the findings reported in this manuscript.

2) The Discussion mentions 'primary visual cortex' (V1) as being an area of high overlap (Figure 4). However the regions appear to be located predominantly within higher visual areas along the lateral wall (rather than within the calcarine sulcus). Please justify or correct this claim.

Thank you for highlighting this, we have corrected this in the revised manuscript:

“Our findings show that spatial overlap was prominent in the temporo-parietal-occipital junction and within higher visual areas along the lateral wall (Figure 4).”

3) As the core brain-behavior finding from the CCA analysis is from Snet PFM, would there be any way to visualize the pattern of spatial overlap associated with the components (along the lines of the figures shown in Figure 4A)? This might help in interpreting the results in the context of prior findings.

Thank you for this excellent suggestion. In the revised manuscript we include a video of overlap associated with the main positive-negative CCA mode (Figure 8—video 1). This video maps directly onto the videos included in our earlier work, and shows the change in overlap as a function of behaviour. We have included these results in the revised manuscript:

“To facilitate the interpretation of these findings in light of our earlier work (Bijsterbosch et al., 2018), we include a video of spatial overlap as a function of the most significant behavioural CCA mode (Figure 8—video 1). Here, we followed the same procedure described previously to generate interpolated video frames for each of the 50 PFM maps, and subsequently summed the grayordinate-wise weights across all 50 maps following normalisation (as for Figure 5A). The resulting video (Figure 8—video 1) shows that spatial overlap systematically varies along with the network-specific spatial variation that we showed previously (Bijsterbosch et al., 2018). Hence, these results show that spatial overlap (and resulting correlation) is a key behaviourally-relevant aspect of spatial information.”

4) The Discussion mentions that ongoing work explores why the brain-behavior relationships are also observed when hard parcellations are used (e.g. the Yeo et al. and Glasser et al. parcellations). The authors speculated that: "it is possible that [hard] parcellation methods are therefore unable to isolate overlap into a distinct parcel (which would allow results to be unbiased, particularly at a sufficiently high dimensionality), and instead parcel boundaries in regions of overlap are determined by the network with the strongest amplitude (or lowest cross-subject variance), leading to mixing of extracted timeseries". Are the authors now able to confirm this speculation? More generally, can the authors explain why the brain-behavior variance explained is so similar when using hard parcellations and when using dual regression approaches (Figure 1A of Bijsterbosch et al., 2018)? Should this be the case if the dual regression methods are subject to these spatial overlap / spatial independence effects, while hard parcellations are not?

Our ongoing work to assess the effect of overlapping functional organization on hard parcellations suggests that the accuracy of temporal network matrix estimates is reduced in the presence of overlap, similarly to standard dual regression approaches. Hence, temporal network matrices obtained from both dual regression and hard parcellation methods are affected by spatial overlap, explaining their comparable associations with behaviour. On a mathematical level, the mechanisms by which spatial overlap influences the temporal network matrix estimates differs fundamentally between the two categories of methods. As shown in this current paper, the ICA-DR approach captures the spatial organisation with good accuracy, but minor errors in spatial estimation (resulting from spatial independence) introduce systematic biases when calculating temporal correlations between extracted timeseries. On the other hand, our preliminary findings show that hard parcellations likely mis-estimate the spatial organisation in the presence of spatial overlap. This affects the network matrices as a result of averaging together timeseries from different functional systems (timeseries mixing). While these are important methodological differences in terms of the mechanisms by which overlap influences network matrix estimation, the effects on brain-behaviour investigations are likely to be very similar. We have revised the wording of the re-submitted manuscript to clarify our preliminary findings:

“Early qualitative observations suggest the presence of complex patterns of partial overlap between multiple extended networks, that cannot easily be modelled by increasing the dimensionality, to allow separation of previously overlapping voxels into one new parcel, and non-overlapping voxels into other parcel(s) (which would allow results to be unbiased, particularly at a sufficiently high dimensionality). As such, our preliminary results suggest that hard parcellations are affected by overlapping functional organisation due to the mis-estimation of spatial nodes and resulting mixing of extracted timeseries (Smith et al., 2011). Further work is needed to test this hypothesis and to develop parcellation methods that address this issue.”